# The Role of Biomimetic Hypoxia on Cancer Cell Behaviour in 3D Models: A Systematic Review

**DOI:** 10.3390/cancers13061334

**Published:** 2021-03-16

**Authors:** Ye Liu, Zahra Mohri, Wissal Alsheikh, Umber Cheema

**Affiliations:** 1UCL Centre for 3D models of Health and Disease, Division of Surgery and Interventional Sciences University College London, London WC1E 6BT, UK; wissal.alsheikh.19@ucl.ac.uk; 2Medical Biosciences, Faculty of Medicine, Imperial College London, London SW7 2BU, UK; z.mohri@imperial.ac.uk

**Keywords:** 3D, cancer models, hypoxia, cancer stem cells, drug resistance, epithelial-to-mesenchymal transition, invasion, migration

## Abstract

**Simple Summary:**

Cancer remains one of the leading causes of death worldwide. The advancements in 3D tumour models provide in vitro test-beds to study cancer growth, metastasis and response to therapy. We conducted this systematic review on existing experimental studies in order to identify and summarize key biomimetic tumour microenvironmental features which affect aspects of cancer biology. The review noted the significance of in vitro hypoxia and 3D tumour models on epithelial to mesenchymal transition, drug resistance, invasion and migration of cancer cells. We highlight the importance of various experimental parameters used in these studies and their subsequent effects on cancer cell behaviour.

**Abstract:**

The development of biomimetic, human tissue models is recognized as being an important step for transitioning in vitro research findings to the native in vivo response. Oftentimes, 2D models lack the necessary complexity to truly recapitulate cellular responses. The introduction of physiological features into 3D models informs us of how each component feature alters specific cellular response. We conducted a systematic review of research papers where the focus was the introduction of key biomimetic features into in vitro models of cancer, including 3D culture and hypoxia. We analysed outcomes from these and compiled our findings into distinct groupings to ascertain which biomimetic parameters correlated with specific responses. We found a number of biomimetic features which primed cancer cells to respond in a manner which matched in vivo response.

## 1. Introduction

Cancer remains one of the leading causes of death despite notable advances in modern medicine [1]. In vitro research has been key to understanding in vivo tumour cellular growth and progression. Since the initial discovery of 2D cell culture techniques in 1907 [2,3], many have used these techniques to study the tumour microenvironment but advances in 3D cell culture and bioengineering techniques have gone further in truly recapitulating in vivo tumour cellular characteristics such as pericellular hypoxia, epithelial-to-mesenchymal transition (EMT) and response to various chemotherapy agents.

Several previous literature reviews of experimental studies involving 2D and 3D in vitro tumour models, the microenvironmental features of different tumour models and drug responses have drawn conclusions that 3D in vitro tumour models mimic physiological characteristics and display biomimetic response to chemotherapy [3,4,5,6,7,8,9,10,11,12]. However what lacks in existing literature is an updated comprehensive systematic analysis of how specific micro-environmental features direct or alter such cellular response.

In this review we have systematically analysed the effect of the addition of biomimetic features, including physiological hypoxia and 3D culture, on cancer cell behaviour. We have focused in particular on cancer cell growth, invasion, EMT and response to therapeutic interventions within biomimetic 3D cultures.

## 2. Materials and Methods

The systematic review was conducted using a number of search engines including “Medline“, “Embase“ and “Web of Science“ with the same initial search terms: “cancer“ and “hypoxia“, then narrowed down further using terms including “2D“ and “3D“ for all databases used. The databases above were all accessed on 28 May 2020. All publications in English language until 28 May 2020 were included for screening. Spelling variations and similar search terms were included for thorough inclusion of existing publications (Table 1).

Inclusion and exclusion criteria were decided and a total of 475 citations were identified through the initial search, excluding duplications, non-full text and non-eligible publications according to the exclusion criteria, 141 publications were deemed eligible for systematic qualitative analysis (Table 2, Figure 1).

Lists of materials, methods, outcomes and conclusions from each publication were recorded via an Excel spreadsheet. To minimize outcome reporting bias, data collection was conducted by two independent investigators, with a third investigator comparing the two data sets for discrepancies and any identified were further analysed by a fourth investigator before final outcome from the specific publication was recorded for the qualitative analysis.

## 3. Results

### 3.1. Level of Hypoxia and Duration

Pathological hypoxia plays an established role in tumour growth, invasion and migration, involving mechanisms such as EMT, angiogenesis and glucose metabolism. Tumour masses of over 500 micrometer diameters display gradients of oxygen concentration, affecting aspects of tumour growth as described above [5]. Therefore, it would come as no surprise that 3D in vitro tumour models will display a degree of intrinsic hypoxia regardless of external hypoxia represented as pO_2_ or partial pressure of oxygen in the overlaying media. The pO_2_ within 3D cultures can drop to 0% causing cell migration away from the core of 3D cultures where this is measured directly within 3D tissue models [13].

In this review a total of 51 publications [14,15,16,17,18,19,20,21,22,23,24,25,26,27,28,29,30,31,32,33,34,35,36,37,38,39,40,41,42,43,44,45,46,47,48,49,50,51,52,53,54,55,56,57,58,59,60,61,62,63,64] utilized various levels of external hypoxia control, primarily through control of the atmospheric oxygen level of the chamber. The remaining studies validated hypoxia through a number of different routes including; internally generated hypoxia within cultures through the use of pO_2_ probes to measure the media and or 3D tumour models for dissolved oxygen; stains such as pimonidazole [23] as a tumour hypoxia marker and the upregulation of Hypoxia-Inducible Factor as an indicator of hypoxia. Publications involving multiple studies with different hypoxia levels, yielded a total of 65 studies. Levels of external hypoxia within these studies ranged from 0.1% to 10% oxygen, with 71% of studies using a hypoxic level of 1% oxygen or less. Control cultures for normoxia were set at either 20% or 21% oxygen with the exception of one study that used 17% oxygen [17]. The hypoxic exposure time for these studies range between 2 h and 70 days with 71% of studies exposing cultures to hypoxic condition for 72 h or less, and 52% of studies using specifically 24 or 48 h.

When directly measuring oxygen in tissues, this is done so by measuring the partial pressure of oxygen (mmHg) and therefore reported levels of tissue oxygenation use this measure. The pO_2_ of arterial blood is a measure of the effectiveness of transfer of oxygen from the atmosphere (air) by the lungs. Once dissolved, this oxygen is measured using mmHg units, and 7.8 mmHg is roughly equivalent to 1% atmospheric oxygen. The majority of studies in this review have not directly measured hypoxia in 3D cultures, but instead utilised hypoxia chambers to control or “set“ the free atmospheric oxygen. Many researchers will rely upon the equilibrium achieved between free atmospheric oxygen of the chamber and dissolved oxygen in the culture media, and thus this was difficult to verify for each study. There are issues around how quickly this process occurs, especially given aspects such as: (i) the constant temperature of the culture; (ii) the opening and closure of hypoxia chambers, which can cause fluctuations in oxygen tension; (iii) an understanding of the oxygen diffusion coefficient of the 3D scaffold materials used in 3D culture and; (iv) the multiple monitoring of the media pO_2_ and the 3D culture pO_2_, which should preferably be done in the core and at the surface [65].

Tissue normoxia, measured as pO_2_, is distinct from atmospheric oxygen concentration, which is 21% in air [66]. Existing literature has shown that normal human tissue-specific oxygen concentration lies between 2 and 6% oxygen, termed physiological hypoxia, with average normal tissue oxygenation not far above that of 3 to 7.4% oxygen [67]. Taking into account the average range of oxygen concentrations within an untreated tumour which is described in existing literature to range between 0.3 to 4.2% oxygen [67], one would expect studies involving in vitro tumour models to incorporate the above factors, but surprisingly nearly a third of the studies still used oxygen levels of 2% or above, which coincides with the average range for normal physiological hypoxia. Furthermore, the half maximum expression of HIF1a and HIF1b was found to be between 1.5 to 2% oxygen, with the maximum expression at 0.5% or less oxygen. Considering the substantial regulatory effects of HIFs on tumour growth and invasion, especially its proangiogenic properties, careful selection of oxygen concentrations in hypoxia studies should be pertinent and taken into account in experimental designs for in vitro tumour cultures to truly recapitulate the in vivo microenvironment (Figure 2).

### 3.2. Scaffold and Non-Scaffold Based Approaches Are Used Equally in 3D Cancer Studies

3D tumour models are considered to better recapitulate the in vivo tumour microenvironment compared to standard 2D cultures. In this review we have identified a total of 37 publications [34,50,57,60,62,64,68,69,70,71,72,73,74,75,76,77,78,79,80,81,82,83,84,85,86,87,88,89,90,91,92,93,94,95,96,97,98] where 3D culture significantly altered cancer cell behaviour towards a more biomimetic response. Eighteen of these studies used 3D scaffold-based cultures and 18 studies used non scaffold-based, cell-generated extra-cellular matrix (ECM) cultures, with a single study using both approaches [73] (Figure 3). Looking further into the scaffold-based cultures, we found that 54% of studies used non-native scaffold materials and 46% using native scaffold materials. Of each of the cohorts above alginate gel was the most used non-native scaffold and collagen for native scaffold material (Figure 4).

These studies used a variety of measurements to determine the extent of biomimicry within in vitro cultures, using a range of criteria including: Genetic expression of angiogenic and matrix remodelling genes, protein synthesis of angiogenic, invasive and matrix remodelling molecules,; morphological basis involving regulators such as expression of hypoxia induced factors, proangiogenic factors, EMT markers, and factors associated with tumour cell stemness; increased IL-8 expression; measured increases in growth of cancer and invasion.

### 3.3. 3D and Hypoxia Enrich Cancer Stem Cell Expression

3D culture and the presence of hypoxia, measured by the various methods outlined in Section 3.2, enhances cancer stem cell (CSC) expression when compared to 2D or normoxic cultures. A total of 14 publications were identified that described measurable changes in various CSC between 2D and 3D in vitro tumour models [19,21,23,48,72,73,74,89,92,99,100,101,102,103]. 13 out of the 14 publications showed increased CSC markers in 3D than 2D cultures and one publication [99] stated the latter. Of the 13 publications describing 3D cultures displaying upregulated CSC markers, the tumour models used were mainly scaffold-based (64%) compared to non-scaffold-based (36%). Furthermore, 67% of those scaffold-based cultures comprised of native scaffold materials rather than non-native materials (33%). Measurable markers of CSC expression vary between each publication, but we identified the most frequently used markers include CD44 [48,72,74,89,100,101,103], OCT4 [19,23,73,99,102], NANOG [23,72,99], SOX2 [21,100], CD133 [48,72,74] and ALDH1A [23,48,74,103]. We noted in one publication the expression of HIF-1 and SOX2 via immunofluorescent staining displayed more pronounced expressions towards the peripheries of the 3D spheroids. The upregulation of these markers were also analysed through RNA sequence analysis and western blots [21].

### 3.4. 3D Models Enhances Measurable EMT in Cancer Cultures, with Scaffold-Based Cultures Playing a Key Role

EMT is a process identified as one of the key features in tumour invasion and metastasis. We identified a total of 19 publications [15,24,50,69,73,74,76,82,84,92,94,99,100,101,102,104,105,106,107] comparing markers of EMT in 2D vs. 3D tumour cultures, with 18 of those showing enhanced measurements of EMT in 3D compared to 2D cultures. In contrast with the generalised cohort of studies concerning biomimicry of in vitro cancer cultures, the studies specifically concerning EMT consisted of a majority of scaffold-based cultures at 67%, and 33% involving non-scaffold-based cultures. This emphasises that we have identified a positive correlation on the presence of high levels of ECM in cancer cultures to the process of EMT (Figure 5).

### 3.5. Culturing Cancer Cells in 3D Enhances Drug Resistance

We identified a total of 42 publications [16,19,22,23,30,41,42,45,49,52,56,59,61,64,68,72,74,80,85,89,100,101,102,103,105,106,108,109,110,111,112,113,114,115,116,117,118,119,120,121,122,123] comparing cell response to chemotherapy and other interventions between 2D and 3D in vitro tumour models, using a variety of anticancer agents and taking into account any publications which involved more than one type of anticancer agents, yielding in total 85 separate findings, including 79 outcomes concerning chemotherapy agents, three involving radiotherapy, and a further five exploring natural compound agents such as orange-peel flavones. The definitions for drug response varied throughout the studies in terms of markers used and duration over which drug responses were measured. The definition of drug response was variable and included measurements of tumour or spheroid size, viability at 3 days, proliferation rates, expression of angiogenetic and metastatic markers pre- and post-drug exposure.

Overall, 85% of experimental outcomes showed 3D tumour cultures displayed increased resistance to anticancer therapies and 15% showed increased sensitivity in 3D when compared with 2D cultures. Of those which showed increased sensitivity in 3D, the therapies tested were predominantly proapoptotic agents, novel targeted therapy, hypoxia activated drugs, radiotherapy or natural compounds (Figure 6).

Furthermore, we investigated the specific response from each cancer type to multiple therapeutic interventions. We categorized the cell lines into 11 organ systems. All cancer types showed a trend of overall increased resistance in 3D compared to 2D in vitro tumour cultures. Most cancer types showed similar responses ranging from 72% to 88% of studies displaying enhanced resistance to chemotherapy in 3D. Studies involving urological and head & neck cancers reported 100% chemo resistant outcome in 3D, whereas prostate cancers and sarcomas showed a significantly lower proportion for chemo resistant outcomes in 3D compared to other cancer types at 57% and 56% prospectively (Figure 7).

We identified 42 publications from our cohort that specifically studied chemotherapy resistance in 3D cultures, and similar numbers of studies used in scaffold-based cultures and non-scaffold-based cultures (53% vs. 47%). On further investigation, the studies where scaffold-based 3D cultures were used, a much greater percentage of studies used native materials (72%) compared to non-native materials (28%). Similar to the cohort of studies concerning biomimicry of tumour culture, collagen and alginate appear to be the most frequently used materials for native and non-native scaffolds. This noteworthy finding could be due to native ECM proteins being deemed more biomimetic as a 3D scaffold, with materials such as collagen used extensively. Collagen type I is found in the dense fibrotic tissue and the tumour microenvironment, and by using this material, studies report the use of native materials to be more consistent between in vitro and in vivo findings (Figure 8).

### 3.6. Invasion vs. Migration Are Distinctly Different but Related Processes in Cancer Growth and Metastasis

When describing cancer progression and metastasis, invasion and migration often go hand in hand and are described as parts of a process facilitating cancer spread. We investigated the evidence within the scope of this review for the definition and measurable markers used for invasion and migration, and whether these were distinct to one or the other process. 16 publications [24,25,28,32,36,47,49,55,62,82,85,89,92,99,112,124] were identified which described the process of invasion or migration or both. Techniques of measuring aspects of migration and invasion involved mostly imaging assays, with a small number of publications conducting quantitative RT-PCR and immunofluorescent staining on various genetic expression and cell surface markers for each of the processes (Table 3).

Migration imaging assays describe the process as the movement of tumour cells from one position to another, for example across porous membranes, movement within the tumour spheroids or out into the surrounding matrix in an in vitro setting. The morphology of cancer invasion is described in a more aggressive manner, with terms such as invasive, spiky projections and invadopodias into the surrounding ECM. Fibronectin coding genes were found to be up-regulated in studies describing cancer cell migration and various EMT markers such as vimentin, E-cadherin, MMP2 and MMP9, which were all associated with and used to measure cellular invasion. Only invasion is described in an in vivo setting, which has been duly noted in multiple publications. Florczyk et al. described the two cellular processes as, “invasion in vivo and migration in vitro by effective degradation of ECM” [89]. Therefore, it would appear that migration is a process exclusively observed and described in the in vitro setting, whereas invasion can be observed and measured in both in vitro and in vivo. We can also draw the conclusion that cancer invasion is heavily associated with EMT because the majority of PCR and immunofluorescent staining markers used to measure invasion are also used for assessing EMT in cancer cells. From the 16 publications included in this review we can infer that both migration and invasion describe the process of movement of cells, but migration refers to movement only and invasion implies additional intrinsic cellular changes and remodelling of ECM components.

## 4. Discussion

It is important to consider the average range of oxygen concentration in physiologically and pathologically hypoxic tissues when conducting in vitro tumour cultures. Normal tissue oxygen saturation is based on pO_2_, and is distinct from ambient atmospheric oxygen, which is 21%. Tissue normoxia, or physiological hypoxia, ranges between 3–7%, thus when conducting comparisons of hypoxic vs normoxic cultures in cancer studies, a more appropriate choice for hypoxia level should be no higher than 2% oxygen and normoxia control samples should be conducted within the “normal” physiological oxygen levels to truly represent the in vivo tumour and stromal microenvironments [67].

To highlight the significance of ranges of hypoxia, over a fifth (22%) of studies included in this review explored tumour hypoxia using 3% oxygen or higher for the hypoxic samples, and furthermore 5% of studies [22,33,47,51] used >7% oxygen which is in fact a higher level than the average normal tissue oxygen concentration. Furthermore, the oxygen level for normoxic control samples used in 51 studies, at atmospheric oxygen concentration (17–21%), are far higher than physiological tissue normoxia encountered in previous literature (3–7%). By conducting studies at higher oxygen levels than that of the average range for pathological hypoxia, one could alter the expression of HIFs. HIFs and its wide range of transcriptional activities, for example upregulation of pro-angiogenic genes such as VEGF and IL8 and downregulation of angiogenic inhibitors such as angiostatin and endostatin, are crucial in tumorigenesis. Hitherto, the oxygen range required for HIF expression and regulation should be taken into account to produce a truly biomimetic in vitro tumour models [43,125]. Although the half maximum expression of HIFs has been documented to occur between 1.5–2% oxygen [67], we must also take into account that in some tumour cells, HIF expression can persist at a much lower oxygen level compared to normal cells due to their constitutive or genetic changes which lead to altered regulation of the expression of HIFs.

Apart from the level of hypoxia involved in these in vitro studies, the length of exposure also plays a vital role for the biomimicry of these tumour models. There is evidence to suggest that acute hypoxia is more likely to contribute to the malignant progression of tumour cells when compared with chronic hypoxia, meaning that an exposure period of 72 h or less used by the majority (71%) of tumour models included in this review is a sufficient period for hypoxic exposure for malignant progression of most cancer cells. HIF1’s activation has been shown to upregulate EMT within 18hrs of exposure to 1% oxygen and upregulation of angiogenesis related markers can be seen within 2 to 4 h [126]. One study we encountered explored cycling hypoxia and reoxygenation in their tumour cultures [61]. Although evidence does show acute hypoxia exposure is sufficient in tumorigenesis, the repeated cycles of hypoxia/reoxygenation have also been linked with increased angiogenesis, and genetic instability due to induction of DNA damage and impaired DNA repair mechanisms [127].

In this review we can deduce that 3D tumour models have shown to be a better representation of the in vivo tumour microenvironment regardless if they are scaffold based or rely upon cell-generated ECM deposition to develop 3D matrix, when compared to 2D cultures. Multicellular spheroids and ECM/scaffold-based tumour cultures have similarities in their shared three-dimensional nature in terms of presence of hypoxic regions and their effects on tumour growth and metastasis. Both scaffold and non-scaffold-based tumour cultures demonstrate that tumour cells play a significant role in the secretion of various ECM proteins such as collagen and fibronectin. There is significant data to suggest that non scaffold based in vitro cancer cell cultures, with an absence of extracellular ECM have a less significant effect on the biomimetic properties of the tumour models in terms of hypoxia driven gene expressions, cell ECM interaction driven growth and invasion, and malignant progression. The role of ECM in facilitating tumour invasion and migration is still not fully understood, however evidence has shown that ECM remodelling and infiltration of ECM is strongly primed by various soluble factors by the infiltrating tumour cells [128]. This would suggest that the indifference between the proportion of scaffold and non-scaffold based studies in this review could be explained by the innate ECM secretion and alteration abilities of the tumour cells.

However, when we analysed the studies exclusively exploring the role of 3D tumour models on EMT, the majority of studies involved scaffold based cultures in comparison with studies looking at biomimicry as a generic theme. ECM components such as collagens, fibronectins, and hyaluronic acid have been shown to play a crucial regulatory role in the EMT signalling pathways. Studies have shown growth factors such as TGF-B, which is present in or sequestered by ECM components, via the TBG-B-ECM axis increases expression of pathways for EMT execution in a dependent manner. Overall, ECM not only provides a structurally supportive role in cancer growth and invasion, but also ECM derived signals provide critical anchor points involved in EMT during the tumorigenesis process and the fine-tuned interaction between the ECM and tumour cells have a vital role in pathogenesis of cancer [129]. Therefore, the use of scaffold based in vitro tumour cultures for the study of EMT specifically is reported to enhance the EMT process given the involvement of ECM in this process. This does not undermine the role of cell-generated ECM in non-scaffold based, however the level of pre-existing ECM has a strong positive correlation with enhanced EMT in tumour models.

CSCs have been isolated and characterised from numerous types of solid cancers such as breast, colorectal, and hepatocellular carcinomas. Their role in tumorigenesis and metastasis has been described in previous literature and their self-renewal, replicative and differentiative abilities via common signalling pathways have been hypothesised to be the basis for metastatic tumour growth [130]. This review highlighted the upregulation of CSC markers in 3D in vitro tumour constructs when compared with 2D, a key feature of cancer evolution, thus reiterating the supporting evidence for enhanced biomimicry features of 3D tumour models compared to 2D cultures.

The preference of native material over non-native material for scaffold based in vitro tumour models in drug resistance studies is the logical approach since the purpose of these experiments are for prediction of in vivo therapy response in an in vitro laboratory setting. Therefore, in order to recapitulate the natural in vivo tumour microenvironment, the choice of natively occurring scaffold material is an obvious preference.

Response to all therapy classes conveyed a trend of increased resistance to therapy in 3D in vitro tumour cultures compared to 2D cultures. Furthermore, five out of six of the chemotherapy classes expressed congruent response of increased resistance to therapy in all experimental setups in 3D compared to 2D. Pro-apoptotic agents stood out showing almost even-handed response. We uncovered that 9/10 experimental setups expressing sensitivity to proapoptotic agents used the drug triapizamine. Tirapazamine, a hypoxia- activated agent, showed increased sensitivity in 3D tumour in all the studies which involved the drug. Due to the presence of intrinsic hypoxic cores in 3D tumour constructs, tirapazamine targets hypoxic regions of the tumour, therefore is highly sensitive in such tumour cultures compared to 2D tumour cultures which lack hypoxic regions [131]. The targeted therapy class displayed a relatively high count of sensitive responses compared to other classes of therapy [56]. As this group includes unclassified chemotherapeutic agents understudied novel therapies and enzyme targets, it is hard to identify the reason of the unusual cellular response to these therapies. Further research is encouraged to promote understanding of these drugs, and their efficacy in vivo, since their anti-tumour effect in more biomimetic 3D models is favourable.

The proportional increase in chemosensitivity of prostate cancer and sarcomas can be accounted for by the multiple specific enzymes targeting novel chemotherapy agents tested in each of the publications compared to other cancer types studied in other publications. When sarcoma cell lines responses to novel targeted therapy DZNep compared to a classical chemotherapy agent cisplatin, all cell lines displayed chemosensitivity to DZNep and resistance to cisplatin [80]. The limited literature regarding these novel chemotherapy agents demonstrates the need for further in depth understanding into these novel chemotherapy agents is required to evaluate the outcomes from studies which involve novel experimental chemo agents.

The increased drug resistance in 3D tumour cultures compared to 2D culture systems is not only owing to its oxygen diffusion gradient. The presence of hypoxia is a well-known factor responsible for drug resistance via several difference mechanisms such as hypoxia induced G0 cell cycle arrest [132]. The increased CSC markers expression should also be considered as a factor in increased drug resistance in 3D cancer cultures as the presence of CSCs have been shown to increase chemoresistance and tumorigenicity [133]. Therefore, cancer chemoresistance should be studied as a multifactorial, multi-causative feature taking into account various key aspects of the tumour microenvironment such as scaffold material, hypoxia level and the involvements of CSCs.

Cell migration in the non-malignant setting usually refers to the orchestrated movement of cells in a particular direction to a specific location [134]. Cell invasion has been defined by various publications with an example from *Nature Research* stating invasion as “the ability of cells to become motile and to navigate through the ECM within a tissue or to infiltrate neighboring tissues“ [135]. Nevertheless, the common theme amongst publications defining cell invasion is the pertinent role it has in cancer progression and metastasis. Many authors describe cell migration to be the first step of cancer invasion into the adjacent surrounding which can be supported by the presence of cellular movement from the 3D tumour spheroids into the surrounding ECM and subsequently forming invasive projections via various cell-ECM interactions [136]. Therefore, the two seemingly distinct cellular processes are closely related when it comes to cancer progression and spread, so one could argue that invasion and migration ought to be investigated as stages of the same process rather than two separate cellular actions. There is also evidence to support the concept of migration being a benign process whereas invasion is closely related to EMT therefore possessing the hallmarks of malignant cellular behaviour.

It is crucial to recognise possible reporting biases which may have resulted during the course of this comprehensive review. Publication, time lag and language biases were noted since this review deemed only full text publications up until 28 May 2020 in English language eligible for analysis. Outcome reporting bias was also taken into account and two independent investigators were involved in the data collection process separately, without influence from external parties. Nevertheless, taking into account of possible biases, this report should still show a relative comprehensive analysis of the topic due to the inclusive search and analysis methods.

## 5. Conclusions

To conclude, this comprehensive systematic review has analysed in depth the published outcomes in existing literature of the role of 3D cell culture techniques and the corresponding physiological hypoxia on cancer cell behaviour, in particular focusing on growth, invasion, EMT and response to therapeutic interventions. Hypoxia level and duration is a crucial consideration when partaking in in vitro tumour hypoxia studies and must reflect the tissue oxygen concentrations. Whether to use scaffold-based tumour models should be carefully considered when exploring the effects of 3D tumour models on EMT and furthermore the choice of scaffolds in drug resistance studies highlights the importance for in vitro tumour models to recapitulate the in vivo tumour microenvironment. Migration and invasion are distinct but related processes both play crucial roles in cancer progression and metastasis. The development of and optimisation of in vitro 3D tumour models using a variety of microenvironments is essential in the study of cancer growth and progression, and advances in cancer treatments.

## Figures and Tables

**Figure 1 cancers-13-01334-f001:**
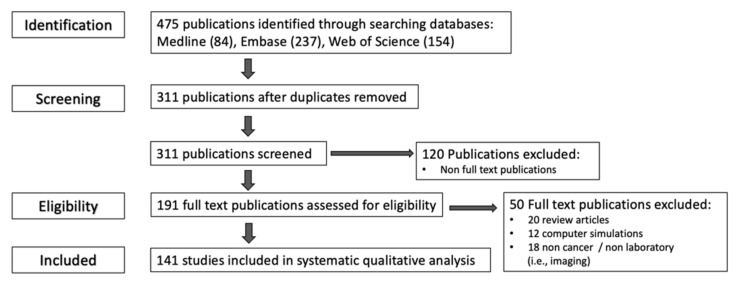
Number of publications excluded at each phase of the publication selection process and reasons for exclusion.

**Figure 2 cancers-13-01334-f002:**
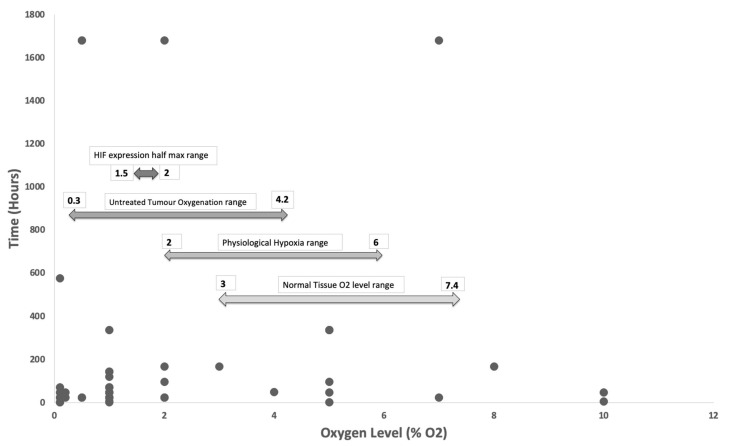
Hypoxia level vs Duration. 71% of studies explored oxygen levels of 1% or below and 71% of studies exposed in vitro cultures for 72 h or less. Arrows demonstrating oxygen range of normal tissue, physiological hypoxia, untreated tumours pathological hypoxia and HIF half maximum expression range. Several studies used oxygen levels outside of the range of physiological tissue hypoxia.

**Figure 3 cancers-13-01334-f003:**
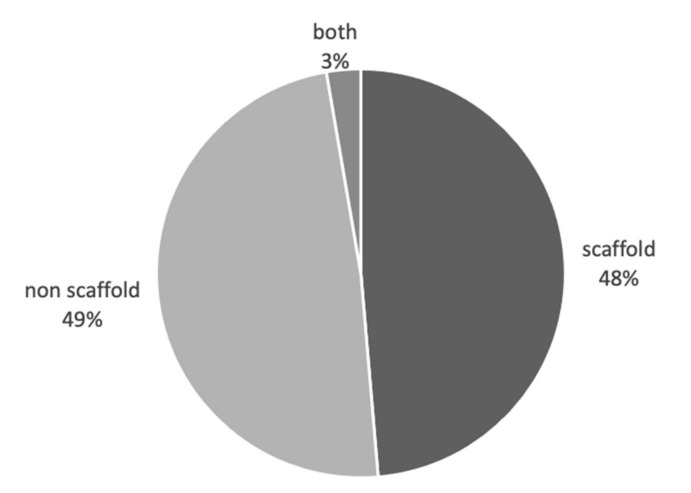
Biomimicry studies: scaffold vs non scaffold vs both. Proportional representation of number of studies using scaffold, non-scaffold-based tumour cultures in biomimicry studies.

**Figure 4 cancers-13-01334-f004:**
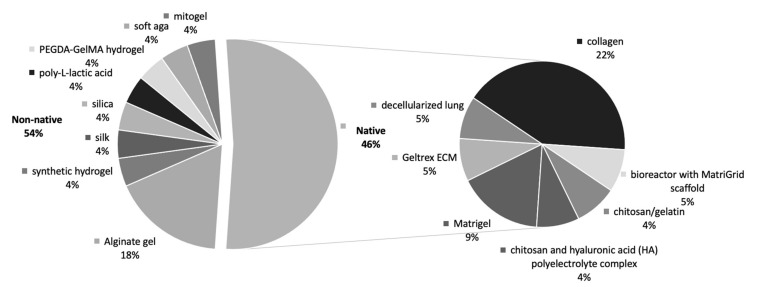
Scaffold Types in biomimicry studies. Type of scaffolds used in scaffold-based studies regarding biomimicry of in vitro cancer cultures. Similar proportion seen of native vs non-native materials for scaffold cultures.

**Figure 5 cancers-13-01334-f005:**
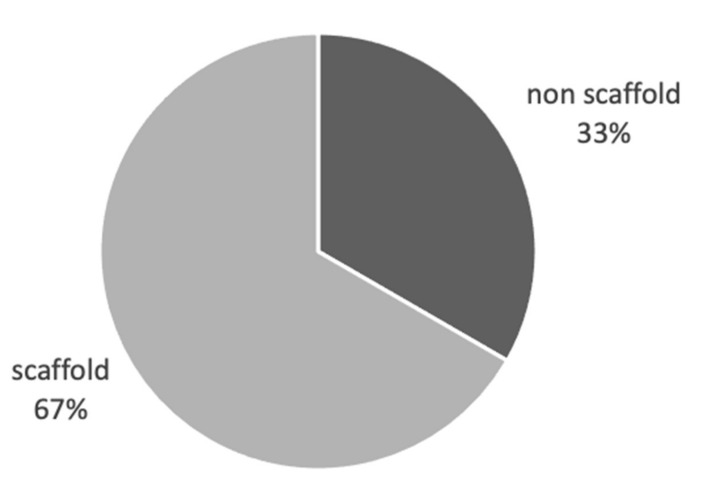
EMT studies: Scaffold vs. non scaffold. Proportional representation of scaffold and non-scaffold based studies exploring EMT, highlighting the importance of ECM in the process.

**Figure 6 cancers-13-01334-f006:**
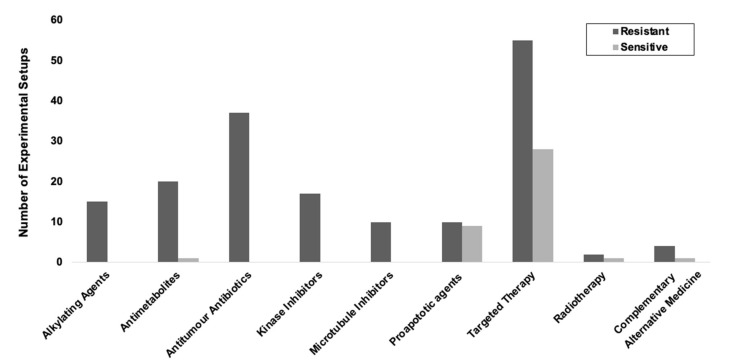
Cancer cell resistance or sensitivity to chemotherapy agents. Response to all therapy classes conveyed a trend of increased resistance to therapy in 3D in vitro tumour cultures compared to 2D cultures. Furthermore, five out of six of the conventional chemotherapy classes expressed congruent response of increased resistance to therapy in all experimental setups in 3D compared to 2D.

**Figure 7 cancers-13-01334-f007:**
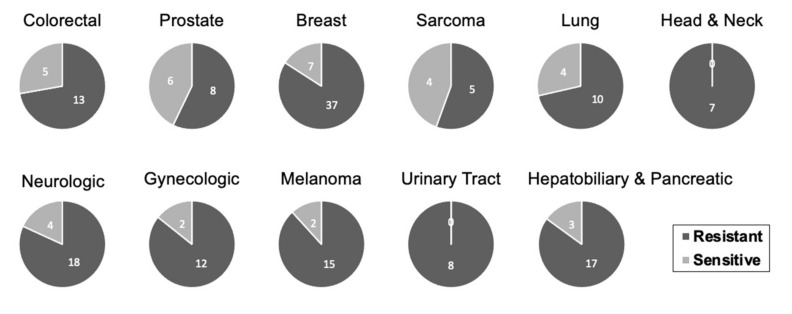
Cancer-specific response to chemotherapy. Each response represents a single experimental outcome. A larger fraction of all cell-lines showed resistance to chemotherapy. Cell-lines that were not frequently studied were categorized under an organ system. Sarcoma includes osteosarcoma, chondrosarcoma, and fibrosarcoma cell-lines. Head and neck cancers include head and neck squamous cell carcinoma, adenoid cystic carcinoma, and ameloblastoma cell-lines. Urinary tract cancers include renal cell carcinoma, renal cell carcinoma bone metastasis, and bladder cancer. Gynecologic malignancies include ovarian and cervical cancer cell-lines. Neurologic cancers include glioblastoma multiforme, neuroblastoma and astrocytoma cell-lines. Hepatobiliary and pancreatic cancers include hepatocellular, biliary tract, and pancreatic cancer cell-lines.

**Figure 8 cancers-13-01334-f008:**
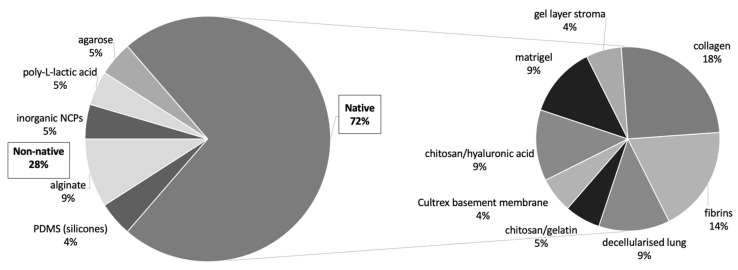
Drug resistance studies: scaffold types. Proportional representation of native and non-native scaffold types used in 3D in vitro tumour models in studies regarding drug resistance with preference towards native scaffold materials over non-native material choices.

**Table 1 cancers-13-01334-t001:** Search terms.

Search Terms	Synonym/Alternative Spellings
Hypoxia	-
Cancer	Neoplasm/tumor/tumour
2D	2 dimension/two dimension
3D	3 dimension/three dimension

**Table 2 cancers-13-01334-t002:** Inclusion and exclusion criteria.

Inclusion Criteria	Exclusion Criteria
All cancer types	Review articles
In vitro and in vivo studies	Computer simulations
2D and 3D studies	Non cancer
Hypoxia studies	Non full text publications

**Table 3 cancers-13-01334-t003:** Measurable markers of migration and invasion. Descriptors of migration are restrict to cell movement, whereas descriptors of invasion include the phrase “aggressive”. (IF- immunofluorescent. PCR- polymerase chain reaction).

Measuring Technique	Migration	Invasion
Imaging (14 publications)	Transwell migration assay across porous membrane. Position of cells within spheroid change. Cell movement out of spheroid. Movement into surrounding gel matrix. Movement of single cell into aggregates. Movement of cells into a scaffold. Motility assay. Movement on ECM/gelatin, dispersion of cells from a solid and ameboid migration pattern.	Invaded cells from spheroid into surrounding collagen/basement membrane/invasion matrix ECM. Boyden chamber invasion assay. Invasive/spiky projections/invadopodias.
qPCR/protein microarray (3 publications)	FAK. c-Src. FN1	MMP-2, MMP-9. TWIST 1. MRTK. AXL. SNAIL1. SNAIL2
IF stain (2 publications)	n/a	vimentin. E-cadherin.
Related to EMT	n/a	Yes (PCR and IF stain markers all notable EMT markers)
Measured in vivo/in vitro	In vitro	in vivo and in vitro

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
