# Peer review of "The Role of Biomimetic Hypoxia on Cancer Cell Behaviour in 3D Models: A Systematic Review"

_cancers, 2021, doi:10.3390/cancers13061334_

Round 1

Reviewer 1 Report

The review by Liu et al….summarizes publications in the field of hypoxia and 3D cancer models. Herein, a number of papers were screened specially from the aspect of the difference of 2D versus 3D hypoxic tumor models. These data were analyzed and listed in different chapters. The article attemps to answer the question :“what in lacking in existing literature is an updated comprehensive systematic analysis of how  specific microenvironmental features control or alter such cellular response”.

Minor revision

Line 9-16: A “simple summary” was given, and additional an abstract. This is unusually, why is a simple summary given?

Line 31 the authors wrote: “ chief causes“  please find an alternative description.

 Line 32: Ref 1 does not work at all. Please correct address!

Line 33  authors wrote  : “Since the initial discovery of 2D cell culture techniques in 1907 [2],”

However, Ref 2 (https://dtp.cancer.gov/timeline/flash/milestones/M4_Nixon.htm)

provide some information on the National Cancer Act of 1971, but there is no information on 2D cell culture techniques in 1907!

However, such an wonderful manuscript as Liverani et al. was not included, although such a paper should meet your criteria? (A biomimetic 3D model of hypoxia-driven cancer progression. Sci Rep. 2019 Aug 22;9(1):12263. doi: 10.1038/s41598-019-48701-4. Are the criteria you have used are correct?

Line 132 authors wrote  :” 3D culture and the presence of hypoxia enhances cancer stem cell (CSC) expression, when compared to 2D or normoxic cultures …A total of 14 publications were identified  that contained measurable changes in various CSC between 2D and 3D in vitro tumour models.[19, 21, 23, 48, 71-73, 88, 91, 98-102]”.

However, the author of reference Ref 21: Blandin, A.F., et al., Hypoxic Environment and Paired Hierarchical 3D and 2D Models of Pediatric H3.3-Mutated Gliomas Recreate the 458 Patient Tumor Complexity. Cancers, 2019. 11(12): p. 26.)

Blandin et al., wrote : “We showed that the expression of the stem cell marker SOX2 was restricted to the periphery of the spheroid, while HIF-1 was expressed from the core to the periphery of the spheroid”.

The statement by the authors Liu et al., in this manuscript is not wrong but it is also not correct; consider the results published by Blandin et al.,.

This means rather a general criticism of this or other reviews. An undifferentiated generalization of data, which is often not always perfect, as I will point out to you for the Ref. 21, generalizes data and creates an inaccurate summary.

I have, of course, tried to find misstatements or misquotes which you might have made, but to your relief and mine, your interpretation and analysis were good for all the examples I selected. That is, I have no doubt that you screened and read the publication you selected very carefully.

line 174 ( response not respoonse) and line 225 (manner not manor) are two of serveral typos, I found.

The discussion is very well written and various problems in the field of 2D / 3D / hypoxia tumor model were discussed very well.

As one example:

Line 256 “explored tumour hypoxia using 3% oxygen or higher for the hypoxic samples, and furthermore 5% of studies used >7% oxygen which is in fact a higher level than the average normal tissue oxygen concentration”

This is a very good point, yet there is no reference for this statement… please add the Literature that support this statement. So that the interested reader can follow your reasoning by looking in the original literature.

Overall, this review is of interest to researchers studying the topic, and much work has been done that culminates in a well-worded discussion.

Best

Author Response

Reviewer 1

Comments and Suggestions for Authors

The review by Liu et al….summarizes publications in the field of hypoxia and 3D cancer models. Herein, a number of papers were screened specially from the aspect of the difference of 2D versus 3D hypoxic tumor models. These data were analyzed and listed in different chapters. The article attemps to answer the question :“what in lacking in existing literature is an updated comprehensive systematic analysis of how  specific microenvironmental features control or alter such cellular response”.

 We would like to thank the reviewers for the helpful constructive comments. We appreciate the detailed revision comments from the reviewer and have made the appropriate changes to our manuscript where necessary. Many thanks.

 Minor revision

Line 9-16: A “simple summary” was given, and additional an abstract. This is unusually, why is a simple summary given?

A simple summary was according from the cancer manuscript format for authors. We tried to comply with the guidelines for manuscript for the journal.

Line 31 the authors wrote: “ chief causes“  please find an alternative description.

We thank the reviewer for this comment and have changed the wording to ‘Cancer remains one of the leading causes of death’ 

 Line 32: Ref 1 does not work at all. Please correct address!

We have updated the URL for ref 1. The webpage updated since our submission but the new link should work now. https://www.ons.gov.uk/peoplepopulationandcommunity/birthsdeathsandmarriages/deaths/bulletins/deathsregistrationsummarytables/2019

Line 33  authors wrote  : “Since the initial discovery of 2D cell culture techniques in 1907 [2],”

However, Ref 2 (https://dtp.cancer.gov/timeline/flash/milestones/M4_Nixon.htm)

provide some information on the National Cancer Act of 1971, but there is no information on 2D cell culture techniques in 1907!

We thank the reviewer for pointing out this issue. We have now updated the reference to reflect the statement.

However, such an wonderful manuscript as Liverani et al. was not included, although such a paper should meet your criteria? (A biomimetic 3D model of hypoxia-driven cancer progression. Sci Rep. 2019 Aug 22;9(1):12263. doi: 10.1038/s41598-019-48701-4. Are the criteria you have used are correct?

 We are also surprised the abovementioned paper did not show up on from our searches of the 3 different databases. We conducted our searches in each database with the same search terms and the list of citations yielded were thoroughly analysed by 2 independent instigators. It is a shame the manuscript mentioned above was but included in the search results but nevertheless the citations included in this review should give a good general representation of the field of studies in this review and the trend of outcomes encountered in these studies.

Line 132 authors wrote  :” 3D culture and the presence of hypoxia enhances cancer stem cell (CSC) expression, when compared to 2D or normoxic cultures …A total of 14 publications were identified  that contained measurable changes in various CSC between 2D and 3D in vitro tumour models.[19, 21, 23, 48, 71-73, 88, 91, 98-102]”.

However, the author of reference Ref 21: Blandin, A.F., et al., Hypoxic Environment and Paired Hierarchical 3D and 2D Models of Pediatric H3.3-Mutated Gliomas Recreate the 458 Patient Tumor Complexity. Cancers, 2019. 11(12): p. 26.)

Blandin et al., wrote : “We showed that the expression of the stem cell marker SOX2 was restricted to the periphery of the spheroid, while HIF-1 was expressed from the core to the periphery of the spheroid”.

The statement by the authors Liu et al., in this manuscript is not wrong but it is also not correct; consider the results published by Blandin et al.,.

This means rather a general criticism of this or other reviews. An undifferentiated generalization of data, which is often not always perfect, as I will point out to you for the Ref. 21, generalizes data and creates an inaccurate summary.

We thank the reviewer for the constructive comments. We understand the reviewer’s concern with generalising the outcome by Blandin et al., in terms of HIF-1 expressed from the periphery of the spheroids. This paper also conducted gene expression analysis in the form of PCR and western blots which we are aware is generalised to the cellular population in 2D and 3D and non-specific to specific regions of the cultures. We are aware of the difficulties in such studies to draw generalised conclusions and we believe this is the difficulty of conducting systematic reviews on exploratory laboratory studies with non-standardised methods. We have added ‘We noted in one publication the expression of HIF-1 and SOX2 via immunofluorescent staining displayed more pronounced expressions towards the peripheries of the 3D spheroids. The upregulation of these markers were also analysed through RNA sequence analysis and western blots. [21]’ to reflect the specific point raised for a true representation of the data without bias.

I have, of course, tried to find misstatements or misquotes which you might have made, but to your relief and mine, your interpretation and analysis were good for all the examples I selected. That is, I have no doubt that you screened and read the publication you selected very carefully.

line 174 ( response not respoonse) and line 225 (manner not manor) are two of serveral typos, I found.

 We have made the appropriate changes to spelling errors.

The discussion is very well written and various problems in the field of 2D / 3D / hypoxia tumor model were discussed very well.

As one example:

Line 256 “explored tumour hypoxia using 3% oxygen or higher for the hypoxic samples, and furthermore 5% of studies used >7% oxygen which is in fact a higher level than the average normal tissue oxygen concentration”

This is a very good point, yet there is no reference for this statement… please add the Literature that support this statement. So that the interested reader can follow your reasoning by looking in the original literature.

  We thank the reviewer for the above comments, and have added references to support the statement as suggested for the studies which used >7% oxygen level for hypoxia cultures.

Overall, this review is of interest to researchers studying the topic, and much work has been done that culminates in a well-worded discussion.

The authors would like to thank the reviewers again for the comprehensive and constructive suggestions, which were all valid points which have been raised. We have made some alterations to the manuscript to reflect those suggestions which we welcome further comments. We appreciate the reviewer’s appreciation of the amount of work which the authors have put into this review article, and we are grateful for the complimentary comments. Many thanks.

Best

Reviewer 2 Report

TO AUTHORS

The manuscript by Liu et.al., aims to provide a comprehensive systematic review of studies employing 3D cancer cell models, focused on hypoxia (conducted up to May 2020). The authors claim to review the significance of in vitro hypoxia and 3D tumour models on biological processes such as EMT, cancer cell invasion and migration, cancer stem cell populations, EMT and drug resistance.

While the idea of a systematic review on this topic is interesting and relevant to the Cancer Research Community, the authors fail to deliver, starting with the scope: the title and abstract are focused on hypoxia, which is just marginally analysed in the manuscript. Plenty of the studies mentioned along the review were not performed in hypoxia, and the authors fail to distinguish them, as well as to convey the impact of hypoxia on cellular processes. Even more, the section describing the hypoxia parameters seems to have misconceptions regarding oxygen concentrations on air/gas, in solution and cellularly.  

Throughout the manuscript the authors fail in providing a comprehensive analysis on how specific microenvironmental features direct or alter tumour cell response; the methodology employed to review the literature did not focus on the biological nor on the technical aspects of each model, and therefore the review does not contribute to enlighten the community on the general topic, nor on specific features. The English language would need to be extensively edited, with special attention to the miss use of specific terms related with the scientific topic covered.  

Specific points:

  1. Line 17: “humanized tissue models” – this is not a commonly used designation in what concerns in vitro models; it is misleading as it refers usually to “humanized mice models”.
  2. Levels of hypoxia and duration:
    1. The authors mention oxygen level expressed generically as percentage. One assumes they are referring to absolute oxygen levels in the gas phase, but it should be stated. From the sentences regarding physiological oxygen levels, it seems that there might be some confusion between oxygen on air and dissolved oxygen. In culture, oxygen solubility depends on liquid temperature and viscosity, and oxygen diffusion depends on the concentration gradient created between the gas phase and the cells (and therefore on the cellular consumption rates, as well as the distance between the gas phase and the target cells and the mixing). Additionally, within the 3D models, mass transfer will depend on the compactness and thickness of the 3D cellular construct. As such, in figure 2:
      1. the representation may lead to misinterpretation, as it includes generically “3D tumour models” so it is most probably considering as similar very different cellular oxygen levels, due to differences in the experimental set-up.
      2. it is not clear in which set-ups the oxygen ranges indicated for normal tissues or physiological hypoxia were determined, but from the manuscript text, it seems that the authors are mentioning indiscriminately oxygen levels on “air” and dissolved oxygen. It should clearly described and corrected when appropriate.
  • It is not clear what the author means with “untreated tumour oxygenation range”.
  1. Section 3.2: the authors sate that “a total of 37 publications where 3D culture significantly altered cancer cell behaviour towards a more biomimetic response”, lines 112-114. Given the large body of literature on 3D cell cancer models, it is surprising that the authors found such a short number of publications falling into this criterium. Are these studies conducted under hypoxia conditions? It is not clear. The authors should state:
    1. which parameters were used to define “a more biomimetic response” (line 113)
    2. if this analysis in this paragraph and the data presented in Figures 3 and 4 only considered hypoxia studies.
  2. Section 3.3 – CSC and hypoxia
    1. Once again, it is not clear which studies were performed under hypoxia or not. From the information provided in section 3.1, one can conclude that a several of the studies mentioned herein were not performed under hypoxia. Therefore, the title of the section should be adjusted, as well as the content. For a relevant literature review, additional aspects of the culture setting and of the biological interrogations need to be presented.
    2. Line 142: Indicate references where CSCs markers have been analysed. ALDH enzyme exists in various isoforms and mentioning just ALDH1A seems quite random as several aldehyde dehydrogenase enzymes or activity are often determined.
  3. Section 3.4: EMT in 3D cell models
    1. In this section there is also no distinction between hypoxia and normoxia studies and their potential effect on EMT
    2. Line 148 - 152: “In contrast with the generalised cohort of studies concerning biomimicry of in vitro cancer cultures, the studies specifically concerning EMT consisted of a majority of scaffold-based cultures at 67%, and 33% involving non-scaffold-based cultures. This emphasises the importance of ECM to this process.”

These sentences must be rephrased, and the entire section remodelled. The authors cannot, by any means conclude on the importance of ECM for EMT processes based on the literature analysis presented. Without information on the type of scaffold (bioactive or inert; natural or synthetic), the 3D culture settings and the identity and characteristics of the tumour cells employed in the EMT analysis, no biological hypothesis or conclusion can be drawn.

  1. Line 158-159 “comparing chemotherapy response between 2D and 3D in vitro tumour models” - this is wrong as examples of other therapeutic approaches besides chemotherapy are given.
  2. lines 162-163 “alternative complementary medicine” is used to refer to “5 exploring complementary alternative medicine agents such as orange-peel flavones”. Natural compounds with pharmacological and therapeutical properties are not considered “alternative medicine”. It would be more appropriate to infer as “natural compounds”.
  3. Table 3 title is not clear (line 225): “Measurable markers of migration and invasion. Descriptions of migration restrict to movements whereas invasion is described in a more aggressive manor”
  4. In line 221-222 and in Table 3, authors refer to RT-qPCR as quantitative techniques to evaluate migration and invasion, which is not correct, most of the genes identified can give clues on EMT processes but do not necessarily translate migration and invasion.
  5. Figure 6:
    1. The Y axis legend is not clear – setups are distinct studies or distinct cell culture models/setting?
    2. “antitumour antibiotics” and “alternative complementary medicines” are not common designations in the area and should be rephrased or explained.
    3. The legend title - “Cancer cellular response to chemotherapy agents” is misleading as the graph does not show cellular responses, but the number of studies using anti-cancer agents. Not only chemotherapeutic but also radiotherapeutic agents are should.
  6. Figure 7: The information has poor biological meaning as several tumour types, e.g., breast cancer are composed by a very heterogeneous group of diseases and many different cell lines present distinct properties; the relevance of the review is severely diminished by the inability to provide relevant biological information, as well as information on the 3D culture settings employed and relevant specifications of the latter.
  7. line 293-297: “There is significant data to suggest that non scaffold based in vitro cancer cell cultures, with an absence of extracellular ECM may not necessarily have a significant effect on the biomimetic properties of the tumour models in terms of hypoxia driven gene expressions, cell ECM interaction driven growth and invasion, and malignant progression.” – hypoxia driven genes can be upregulated in non-scaffold based methods; the absence of scaffolds does not mean the absence of ECM, there is no evidence presented in what refers hypoxia..
  8. Line 309: TGF-β is not secreted by ECM components.
  9. Line 314-316: “the use of scaffold based in vitro tumour cultures in the study of EMT in specific rather than non-scaffold based models would appear to the superior choice in light of the involvement of ECM in the EMT process.”- non-scaffold models can also exhibit ECM therefore this aspect should be taken into account.

Author Response

Reviewer 2 : hypoxia review

Open Review

English language and style

(x) Extensive editing of English language and style required
( ) Moderate English changes required
( ) English language and style are fine/minor spell check required
( ) I don't feel qualified to judge about the English language and style

Is the work a significant contribution to the field?

Is the work well organized and comprehensively described?

Is the work scientifically sound and not misleading?

Are there appropriate and adequate references to related and previous work?

Is the English used correct and readable?

Comments and Suggestions for Authors

TO AUTHORS

The manuscript by Liu et.al., aims to provide a comprehensive systematic review of studies employing 3D cancer cell models, focused on hypoxia (conducted up to May 2020). The authors claim to review the significance of in vitro hypoxia and 3D tumour models on biological processes such as EMT, cancer cell invasion and migration, cancer stem cell populations, EMT and drug resistance.

While the idea of a systematic review on this topic is interesting and relevant to the Cancer Research Community, the authors fail to deliver, starting with the scope: the title and abstract are focused on hypoxia, which is just marginally analysed in the manuscript. Plenty of the studies mentioned along the review were not performed in hypoxia, and the authors fail to distinguish them, as well as to convey the impact of hypoxia on cellular processes. Even more, the section describing the hypoxia parameters seems to have misconceptions regarding oxygen concentrations on air/gas, in solution and cellularly.  

Throughout the manuscript the authors fail in providing a comprehensive analysis on how specific microenvironmental features direct or alter tumour cell response; the methodology employed to review the literature did not focus on the biological nor on the technical aspects of each model, and therefore the review does not contribute to enlighten the community on the general topic, nor on specific features. The English language would need to be extensively edited, with special attention to the miss use of specific terms related with the scientific topic covered.  

We would like to thank the reviewer for their comments.  The rationale behind performing this systematic review is a lack of such a review in current literature. This is no coincidence, since it is extremely difficult to do a systematic review where there are unique experimental set-ups, no consistent primary and secondary outcomes, huge variations in the perceived notion of hypoxia, both level, duration and method of measurement (outlined in figure 2), and large variation in what was measured. We have therefore spent the last 12 months rigorously collecting any directly comparable data.

Hypoxia is indeed a search term we looked for, however upon detailed reading it was clear that there was no consensus on how this is assessed. We identified numerous ‘modes’ of hypoxia validation; (i) the use of a hypoxia chamber to control the ambient air O2- with very limited studies simultaneously monitoring the oxygen level in the media (i.e. dissolved oxygen) to ensure or establish the rate of transfer; (ii) use of pO2 probes to measure the media and or 3D tissue model for dissolved oxygen; (iii) stains such as pimonidazole as a tumour hypoxia marker; (iv) upregulation of Hypoxia-inducible factor as an indicator of hypoxia. This has made the direct comparison of hypoxia extremely difficult, however we still feel it relevant to start draw attention to this within the wider research community. We have added this detail

We have been careful to draw clear conclusions. The first is there is no clear consensus on what constitutes hypoxia within the field. Figure 2 clearly demonstrates that the level of hypoxia (atmospheric in the surrounding air) and the duration of this hypoxia varies, significantly so, and therefore making is considerably difficult to draw conclusions. However even with this variability, we have tried to draw out general trends documented.

Due to the large volume of papers assessed, by 2 independent assessors, using a strict systematic review criteria we have outlined our major findings. We have analysed the data from a non-biased perspective and collated findings in extensive spreadsheet to look for commonalities. Therefore unlike a standard review, we have truly captured trends in this field, without prejudice.

We have personal experience of measuring (using fibre-optic probes) hypoxia levels within 3D tissues, and deriving oxygen diffusion coefficients in different 3D matrices, so we are aware of the complexities, please see references below. However, the majority of papers do not provide extensive details of methods of O2 measurement or consistent temperature monitoring, in fact most studies simply put their cultures into oxygen chambers without internal monitoring. As this was a systematic review, we trusted that previous peer review were rigorous.

Specific points:

  1. Line 17: “humanized tissue models” – this is not a commonly used designation in what concerns in vitro models; it is misleading as it refers usually to “humanized mice models”.

We accept that this may be misleading and have therefore changed this to human tissue models.

  1. Levels of hypoxia and duration:
    1. The authors mention oxygen level expressed generically as percentage. One assumes they are referring to absolute oxygen levels in the gas phase, but it should be stated. From the sentences regarding physiological oxygen levels, it seems that there might be some confusion between oxygen on air and dissolved oxygen. In culture, oxygen solubility depends on liquid temperature and viscosity, and oxygen diffusion depends on the concentration gradient created between the gas phase and the cells (and therefore on the cellular consumption rates, as well as the distance between the gas phase and the target cells and the mixing). Additionally, within the 3D models, mass transfer will depend on the compactness and thickness of the 3D cellular construct. As such, in figure 2:
      1. the representation may lead to misinterpretation, as it includes generically “3D tumour models” so it is most probably considering as similar very different cellular oxygen levels, due to differences in the experimental set-up.
      2. it is not clear in which set-ups the oxygen ranges indicated for normal tissues or physiological hypoxia were determined, but from the manuscript text, it seems that the authors are mentioning indiscriminately oxygen levels on “air” and dissolved oxygen. It should clearly described and corrected when appropriate.

The reviewer is correct. Many studies only used a hypoxia incubator in which to culture 3D tumour models, as such, very few studies looked at the heterogenous distribution of oxygen within 3D cultures, thus ignoring the difference between free atmospheric oxygen and dissolved oxygen.

We have added text to clarify this and describe the limitations we faced when assessing this criteria.

“When directly measuring oxygen in tissues, this is done so by measuring the partial pressure of oxygen (mmHg). And therefore reported levels of tissue oxygenation use this measure. The pO2 of arterial blood is a measure of the effectiveness of transfer of oxygen from the atmosphere (air) by the lungs. Once dissolved, this oxygen is measured using mmHg units, and 7.8mmHg is roughly equivalent to 1% atmospheric oxygen. The majority of studies in this review have not directly measured hypoxia in 3D cultures, but instead utilised hypoxia chambers to control or ‘set’ the free atmospheric oxygen. Many researchers will rely upon the equilibrium achieved between free atmospheric oxygen of the chamber and dissolved oxygen in the culture media, and thus this was difficult to verify for each study. There are issues around how quickly this process occurs, especially given aspects such as (i) the constant temperature of the culture; (ii) the opening and closure of hypoxia chambers, which can cause fluctuations in oxygen tension; (iii) an understanding of the oxygen diffusion coefficient of the 3D scaffold materials used in 3D culture and ;(iv) the multiple monitoring of the media pO2 and the 3D culture pO2, which should preferably done in the core and at the surface (Cheema et al. 2008).”

Cheema, U. Brown, R.A. Alp, B. MacRobert, A.J. (2008). Spatially defined oxygen gradients and VEGF expression in an engineered 3D cell model. Cell. Mol. Life Sci. 65(1): 177-186

It is not clear what the author means with “untreated tumour oxygenation range”.

  1. Section 3.2: the authors sate that “a total of 37 publications where 3D culture significantly altered cancer cell behaviour towards a more biomimetic response”, lines 112-114. Given the large body of literature on 3D cell cancer models, it is surprising that the authors found such a short number of publications falling into this criterium.

‘Untreated tumour oxygenation range’ refers to the oxygen concentration within the untreated tumours. It is a range of oxygen levels due to the variety of different types of solids tumours exhibiting a range of oxygen concentrations as we would expect from different cancers. The comprehensive list of tumour oxygen levels can be found in the paper which we referenced in our list of citations (reference number 67). We also found this to be a small number. This systematic review revealed that even when the keyword of ‘hypoxia’ was used with 3D and in vitro studies, we only received 451 hits, by applying our specific criteria this was further reduced to 141 full papers. We were equally surprised that given the thousands of papers on 3D models, the papers where external hypoxia is controlled, or internal hypoxia is measured are relatively small.

Are these studies conducted under hypoxia conditions? It is not clear.

We have discussed this point. All studies we identified employed the use of hypoxia as it was in their keywords. Some groups utilised hypoxia chambers, and the levels of atmospheric oxygen level (measured as percentage) were used to describe the oxygen level of the chamber. However some have less detail in. Relatively few studies measure the internal pO2 of either the media or the 3D model itself. But because hypoxia was somehow detected in the study, we included it. Pg 3.

We have added the following to clarify this  Section 3.1, para 2:

In this review a total of 51 publications [14-64] utilized various levels of external hypoxia control, primarily through control of the atmospheric oxygen level of the chamber. The remaining studies validated hypoxia through a number of different routes including; internally generated hypoxia within cultures through the use of pO2 probes to measure the media and or 3D tumour models for dissolved oxygen; stains such as pimonidazole as a tumour hypoxia marker and the upregulation of Hypoxia-Inducible Factor as an indicator of hypoxia.

Publications

The authors should state:

    1. which parameters were used to define “a more biomimetic response” (line 113) if this analysis in this paragraph and the data presented in Figures 3 and 4 only considered hypoxia studies.

These were mentioned briefly on page 6, line 2. Specifically we looked at specific statements authors made about their research- backed up with experimental data including:  

Increase in invasion

increased growth in 3D

increased growth, hif, vegf bFGF in 3D

increased HIF and VEGf expression in 3D even under hypoxic conditions

increased sten cell, EMT, hypoxia, Multipotency markers in 3D

Increased stem and Emt markers in 3D

Increased protein and gene for ECM deposition and matrix-remodelling

increased EMT, altered stemness and proliferation

more 'tumour' like in 3D

more sensitivity in 3D

more biomemetic in 3D

more tumorigenic and angiogenic

more invasive, increased expression of type II collagen in 3D

more biomemetic in 3D

Hypoxia induced dormancy

more biomemetic in 3D

more biomemetic in 3D

Increasing ECM stiffness increases malignancy

Hypoxia increases growth

3D simulasted invasion at 21 days

EPO increases drug resistance in 3D

monolayers: flat shapes, collagen: seemed to interact with each other. Matrigel-seeded cells: round morphologies and did not proliferate 

3D has more biomemtic gene expression

Different genes were responsible for growth in 2D and 3D

3D has more biomimetic gene expression

3D expressed IL-8 and VEGF similar to in-vivo cancer

static model had more metabolic activity (possibly due to hypoxia)

EMT Markers in 3D + mets in in some cell lines in 3D

3D more angiogenic properties, similar to in vivo

Hypoxic core, with increased angiogenesis, and increased EMT and decreased CD 44+

phenotype of cells cultured in the 3D platforms more closely resembles what is seen in vivo compared with the 2D counterparts ratio of mt DNA/nDNA in 3D lower than in 2D

3D showed the biomimetic result. Drugs tested 5FU and Paclitaxil

Comparing tyrosine kinase inhibitor sensitive and resistant cell lines in 2D and 3D. 3D followed similar pattern of sensitivity  to in vivo.

3D response similar to in vivo

more biomemetic in 3D

hypoxia potentiates efflux

We have added some of these extra terms to page 6, line 2 (highlighted):

These studies used a variety of measurements to determine the extent of biomimicry within in vitro cultures, using a range of criteria including:  genetic expression of angiogenic and matrix remodelling genes, protein synthesis of angiogenic, invasive and matrix remodelling molecules,; morphological basis involving regulators such as expression of hypoxia induced factors, proangiogenic factors, EMT markers, and factors associated with tumour cell stemness; increased IL-8 expression; measured increases in growth of cancer and invasion.

Furthermore, If requested we are happy to provide our multiple excel spreadsheets going over each parameter.

  1. Section 3.3 – CSC and hypoxia
    1. Once again, it is not clear which studies were performed under hypoxia or not. From the information provided in section 3.1, one can conclude that a several of the studies mentioned herein were not performed under hypoxia. Therefore, the title of the section should be adjusted, as well as the content. For a relevant literature review, additional aspects of the culture setting and of the biological interrogations need to be presented.

We have now clarified the text to explain the various methods used to validate the term ‘hypoxia’  by authors of the research papers selected. These include stains, gene and protein markers. As such, we feel that the subtitle is appropriate. We have changed the text to clarify this:

3D culture and the presence of hypoxia, measured by the various methods outlined in section 3.2, enhances cancer stem cell (CSC) expression when compared to 2D…..

    1. Line 142: Indicate references where CSCs markers have been analysed. ALDH enzyme exists in various isoforms and mentioning just ALDH1A seems quite random as several aldehyde dehydrogenase enzymes or activity are often determined.

We have added the references to the molecules. In terms of the reference for ALDH1A, we wished to be accurate in our reporting of other authors work and they specifically tested this isoform.

  1. Section 3.4: EMT in 3D cell models
    1. In this section there is also no distinction between hypoxia and normoxia studies and their potential effect on EMT

All studies that we reported where there was an increase in EMT were reported in hypoxia. That was the basic criteria we used. As previously discussed, the range if hypoxia varied, and the measure used to validate hypoxia varied from study to study. We have discussed this at length in the discussion:

“To highlight the significance of ranges of hypoxia, over a fifth (22%) of studies included in this review explored tumour hypoxia using 3% oxygen or higher for the hypoxic samples, and furthermore 5% of studies [22, 33, 47, 51] used >7% oxygen which is in fact a higher level than the average normal tissue oxygen concentration. Furthermore, the oxygen level for normoxic control samples used in 51 studies, at atmospheric oxygen concentration (17-21%), are far higher than physiological tissue normoxia encountered in previous literature (3-7%). By conducting studies at higher oxygen levels than that of the average range for pathological hypoxia, one could alter the expression of HIFs. HIFs and its wide range of transcriptional activities, for example upregulation of pro-angiogenic genes such as VEGF and IL8 and downregulation of angiogenic inhibitors such as angiostatin and endostatin, are crucial in tumorigenesis. Hitherto, the oxygen range required for HIF expression and regulation should be taken into account to produce a truly biomimetic in vitro tumour models.[43, 125] Although the half maximum expression of HIFs has been documented to occur between 1.5-2% oxygen,[67] we must also take into account that in some tumour cells, HIF expression can persist at a much lower oxygen level compared to normal cells due to their constitutive or genetic changes which lead to altered regulation of the expression of HIFs.

Apart from the level of hypoxia involved in these in vitro studies, the length of exposure also plays a vital role for the biomimicry of these tumour models. There is evidence to suggest that acute hypoxia is more likely to contribute to the malignant progression of tumour cells when compared with chronic hypoxia, meaning that an exposure period of 72 hours or less used by the majority (71%) of tumour models included in this review is a sufficient period for hypoxic exposure for malignant progression of most cancer cells. HIF1’s activation has been shown to upregulate EMT within 18hrs of exposure to 1% oxygen and upregulation of angiogenesis related markers can be seen within 2 to 4 hours [126]. One study we encountered explored cycling hypoxia and reoxygenation in their tumour cultures [61]. Although evidence does show acute hypoxia exposure is sufficient in tumorigenesis, the repeated cycles of hypoxia/reoxygenation have also been linked with increased angiogenesis, and genetic instability due to induction of DNA damage and impaired DNA repair mechanisms.[127] “

    1. Line 148 - 152: “In contrast with the generalised cohort of studies concerning biomimicry of in vitro cancer cultures, the studies specifically concerning EMT consisted of a majority of scaffold-based cultures at 67%, and 33% involving non-scaffold-based cultures. This emphasises the importance of ECM to this process.”

These sentences must be rephrased, and the entire section remodelled. The authors cannot, by any means conclude on the importance of ECM for EMT processes based on the literature analysis presented. Without information on the type of scaffold (bioactive or inert; natural or synthetic), the 3D culture settings and the identity and characteristics of the tumour cells employed in the EMT analysis, no biological hypothesis or conclusion can be drawn.

We have re-stated our findings to reflect only the positive correlation that we have identified, and been careful to not assert this as a tested biological phenomenon.

This emphasises that we have identified a positive correlation on the presence of high levels of ECM in cancer cultures to the process of EMT.

  1. Line 158-159 “comparing chemotherapy response between 2D and 3D in vitro tumour models” - this is wrong as examples of other therapeutic approaches besides chemotherapy are given.

We have modified this statement:

cell response to chemotherapy and other interventions

  1. lines 162-163 “alternative complementary medicine” is used to refer to “5 exploring complementary alternative medicine agents such as orange-peel flavones”. Natural compounds with pharmacological and therapeutical properties are not considered “alternative medicine”. It would be more appropriate to infer as “natural compounds”.

We agree natural compounds is a more appropriate term. This has been changed multiple times throughout the text.

  1. Table 3 title is not clear (line 225): “Measurable markers of migration and invasion. Descriptions of migration restrict to movements whereas invasion is described in a more aggressive manor”

“Measurable markers of migration and invasion. Descriptors of migration are restrict to cell movement,  whereas descriptors of invasion include the phrase ‘aggressive’ ”

  1. In line 221-222 and in Table 3, authors refer to RT-qPCR as quantitative techniques to evaluate migration and invasion, which is not correct, most of the genes identified can give clues on EMT processes but do not necessarily translate migration and invasion.

We have raised this point within the discussion. The PCR markers for invasion are accurate, including MMP2 and MMP9. We have noted that these markers have been identified by authors to represent the processes on invasion and migrations. Please see citations regarding the studies quoted.

  1. Figure 6:
    1. The Y axis legend is not clear – setups are distinct studies or distinct cell culture models/setting?
    2. “antitumour antibiotics” and “alternative complementary medicines” are not common designations in the area and should be rephrased or explained.

‘Antitumour antibiotics’ has long been a well-established class of anticancer drug class in the field of oncology with well-known and commonly used examples in current medical practice including doxorubicin and bleomycin. ‘Alternative complementary medicines’ as forementioned we have renamed ‘natural compounds’ but this class of antitumour drugs is a novel class which is still under stages of in vitro studies therefore the descriptions of this class of drugs has a variety of descriptions in existing literature.

    1. The legend title - “Cancer cellular response to chemotherapy agents” is misleading as the graph does not show cellular responses, but the number of studies using anti-cancer agents. Not only chemotherapeutic but also radiotherapeutic agents are should.

We have altered the legend to the following.

“Cancer cell resistance or sensitivity to chemotherapy agents”

We did in fact check whether the studies reported resistance to treatment or sensitivity, and subsequently, we have reported this finding.

  1. Figure 7: The information has poor biological meaning as several tumour types, e.g., breast cancer are composed by a very heterogeneous group of diseases and many different cell lines present distinct properties; the relevance of the review is severely diminished by the inability to provide relevant biological information, as well as information on the 3D culture settings employed and relevant specifications of the latter.

We appreciate the referees concerns. However this review does not aim to conduct in depth analysis of biological meaning. Our main message was that where 3D biomimetic models are used, with hypoxia implicated either through control via a hypoxia chamber, or generated within the 3D model, an increase in drug sensitivity is noted. We believe this to be an important finding and a noteworthy discussion point. In a field where drug efficacy does not match in animal models and human response, using biomimetic models offers an alternative. The more evidence we generate around this will inform our practises as a field.

  1. line 293-297: “There is significant data to suggest that non scaffold based in vitro cancer cell cultures, with an absence of extracellular ECM may not necessarily have a significant effect on the biomimetic properties of the tumour models in terms of hypoxia driven gene expressions, cell ECM interaction driven growth and invasion, and malignant progression.” – hypoxia driven genes can be upregulated in non-scaffold based methods; the absence of scaffolds does not mean the absence of ECM, there is no evidence presented in what refers hypoxia.

We have been careful to interpret data we analysed from the papers we read. There is a correlation with scaffold based in vitro cancer cultures enhancing specific cancer behaviours. We have therefore not definitively stated that non-scaffold based cultures are not representative, rather that the responses, for example in hypoxia-driven genes, are less noted where non-scaffold based approaches are taken. We have nevertheless re-stated this to make this more clear:

There is significant data to suggest that non scaffold based in vitro cancer cell cultures, with an absence of extracellular ECM have a less significant effect on the biomimetic properties of the tumour models in terms of hypoxia driven gene expressions, cell ECM interaction driven growth and invasion, and malignant progression

  1. Line 309: TGF-β is not secreted by ECM components.

We are sorry for this- it was meant to read ‘sequestered’. We have corrected this.

  1. Line 314-316: “the use of scaffold based in vitro tumour cultures in the study of EMT in specific rather than non-scaffold based models would appear to the superior choice in light of the involvement of ECM in the EMT process.”- non-scaffold models can also exhibit ECM therefore this aspect should be taken into account.

Thank you for this comment- we have considered this and re-written this section:

Therefore, the use of scaffold based in vitro tumour cultures for the study of EMT  specifically is reported to enhance the EMT process given the involvement of ECM in this process.  This does not undermine the role of cell-generated ECM in non-scaffold based, however the level of pre-existing ECM has a strong positive correlation with enhanced EMT in tumour models. 

Submission Date

27 January 2021

Date of this review

22 Feb 2021 23:40:52

Best

Reviewer 3 Report

Liu and colleagues used various search engines as Medline, Embase and Web of Science to analyze the published outcomes of 2D and 3D culturing cancer cells and in vitro hypoxia on various aspects of cancer biology such as EMT, cancer stem cells, migration/invasion and drug resistance. The rationale behind the paper is very interesting with important fundamental and therapeutic clinical implications. In general, the authors provide a well-written and useful review. The presented work has applicability to researchers working on various aspects of the hypoxic microenvironment and cancer progression.

Comment #1

The authors should clearly establish the link between one of the main goals of the manuscript, which is the study of hypoxia in 2D and 3D cellular models and the result section on invasion and migration.

Comment #2:

Please indicate the exact period of time for which the publication search was performed. The start date is clearly indicated, but what is the exact end date?

Comment #3

Figures should be in color for better readability

Comment #4

Lines #269-272: please add a reference

Comment #5

Few distracting typos, the entire manuscript must be carefully examined.

Author Response

Reviewer 3

Comments and Suggestions for Authors

Liu and colleagues used various search engines as Medline, Embase and Web of Science to analyze the published outcomes of 2D and 3D culturing cancer cells and in vitro hypoxia on various aspects of cancer biology such as EMT, cancer stem cells, migration/invasion and drug resistance. The rationale behind the paper is very interesting with important fundamental and therapeutic clinical implications. In general, the authors provide a well-written and useful review. The presented work has applicability to researchers working on various aspects of the hypoxic microenvironment and cancer progression.

Comment #1

The authors should clearly establish the link between one of the main goals of the manuscript, which is the study of hypoxia in 2D and 3D cellular models and the result section on invasion and migration.

The authors would like to thank the reviewers for the helpful and constructive comments and suggestions. We have clearly outlined the rationale behind this review article and our aims and objectives which is to draw conclusions of the effects of hypoxia and 3D in vitro tumour constructs on the process of tumour migration and invasion, specifically epithelial to mesenchymal transition in the existing literature up to the date of our database search.

Comment #2:

Please indicate the exact period of time for which the publication search was performed. The start date is clearly indicated, but what is the exact end date?

We have included in our screening process all publications which were published up to the date of our database search on the 28th May 2020 from the three web databases we used. This can be found in the discussion, final paragraph page 13.

Comment #3

Figures should be in colour for better readability

We will provide these for the online version.

Comment #4

Lines #269-272: please add a reference

We thank the reviewer for the above comment, and have added a reference to reflect the statement mentioned in the above lines.  

Comment #5

Few distracting typos, the entire manuscript must be carefully examined.

We have conducted a spelling and grammar check and corrected typos. We are sorry for this.

Submission Date

27 January 2021

Best
